# Potential Mechanisms of Gut-Derived Extracellular Vesicle Participation in Glucose and Lipid Homeostasis

**DOI:** 10.3390/genes13111964

**Published:** 2022-10-28

**Authors:** Tiange Feng, Weizhen Zhang, Ziru Li

**Affiliations:** 1Department of Physiology and Pathophysiology, Peking University Health Science Center, Beijing 100191, China; 2Department of Surgery, University of Michigan Medical Center, Ann Arbor, MI 48109, USA; 3MaineHealth Institute for Research, MaineHealth, Scarborough, ME 04074, USA

**Keywords:** extracellular vesicles, intestine, microbiota, glucose metabolism, lipid metabolism

## Abstract

The intestine participates in the regulation of glucose and lipid metabolism in multiple facets. It is the major site of nutrient digestion and absorption, provides the interface as well as docking locus for gut microbiota, and harbors hormone-producing cells scattered throughout the gut epithelium. Intestinal extracellular vesicles are known to influence the local immune response, whereas their roles in glucose and lipid homeostasis have barely been explored. Hence, this current review summarizes the latest knowledge of cargo substances detected in intestinal extracellular vesicles, and connects these molecules with the fine-tuning regulation of glucose and lipid metabolism in liver, muscle, pancreas, and adipose tissue.

## 1. Introduction

Extracellular vesicles (EVs) are a heterogeneous group of membrane-enclosed structures ranging from 30 to 500 nm in diameter that comprise exosomes, ectosomes, apoptotic bodies, outer membrane vesicles, microvesicles, microparticles, and other EV subsets [1,2]. They are secreted from a broad spectrum of cells, transported in biological fluids, and regarded as a mediator for cells, tissues, and organs to communicate with each other [2]. Various categories of bioactive molecules, including nucleic acids, proteins, lipids, and metabolites have been found encapsulated in EVs [3]. Through their delivery, exchange of genetic and metabolic information was accomplished between neighboring or distant cells. For instance, EVs secreted by adipose tissue can be internalized by remote organs such as the liver, skeletal muscle, pancreas, and brain, which provoked complex changes in the energy metabolism network [4].

Gut hormones and microbiota have been extensively reported to influence whole-body glucose and lipid metabolism [5,6]; however, the roles of gastrointestinal EVs in energy metabolism are still undelineated. Gut-derived EVs mainly derive from three origins, including the intestinal mucosa, gut microbiota, and intestinal immune cells. EVs from the first two sources affect a wide range of biological processes in local or remote cells, while those produced by immunocytes mainly participate in regulation of local immunity responses [7]. To gain more insight on the roles of intestinal EVs on glucose and lipid metabolism, in current review we focus on a discussion about the intestinal mucosa and microbiome-derived EVs.

EVs produced by intestinal mucosal cells contain lipids, microRNA, double-strand DNA (dsDNA), and proteins [8]. Recipient cells undergo membrane fusion with EVs and receive the substances delivered from the parent cells. Outer membrane vesicles (OMVs) are the major type of EVs derived from gut microbiota, and are generated via the vesiculation of bacterial outer membrane during the growth process. OMVs are composed of a phospholipid bilayer with periplasmic proteins, and encapsulate microbial DNAs [9] and bacterial membrane components [10,11,12].

Given that EVs can be transported into blood and mediate interorgan crosstalk over a long distance, and the gut senses energy homeostasis fluctuations, the impact of gut-associated EVs on the physiological and pathological processes of other tissues is noteworthy. In this review, we summarize studies on the gut-associated EV cargoes and the roles of their EV contents in glucose and lipid metabolism. Note that most of the studies either revealed the connection between gut and energy metabolism or identified gastrointestinal cell-derived EVs and their contents, but there is still a lack of direct evidence on how EVs mediate the gastrointestinal effects on glucose or lipid metabolism. Our review reveals the mechanisms that may contribute to the connection between gut-derived EVs and energy metabolism.

## 2. Gastrointestinal EV-Containing Substances Are Associated with Glucose and Lipid Metabolism

EVs derived from intestinal mucosa, organoids, or cell lines have been reported to contain dsDNA, microRNAs, cytokines, enzymes, and other signaling molecules. For example, EVs originated from intestinal epithelial cells (IECs) contain immunomodulatory molecules such as major histocompatibility complex (MHC) I and II [8], intestine-specific marker A33 antigen [13], and various microRNAs that are associated with glucose and lipid metabolism. We list some examples below:

### 2.1. EV-Nucleic Acids (microRNA and DNA)

Of note, microRNAs may have multiple target genes; therefore, it is likely that their overall effects involve or interact with different molecules or pathways. Manipulation of particular microRNA may lead to controversial outcomes between studies due to the complex network, which is a natural challenge in the field.

***MicroRNA Let-7 impairs glucose and lipid metabolism.*** In a recent study, Zhang et al. [14] found EVs derived from normal intestinal organoids carried higher concentrations of let7c-5p microRNA compared with that of EVs from organoids treated with morphine, which is associated with worse outcomes of inflammatory diseases, including inflammatory bowel disease (IBD) and sepsis. Let-7 microRNA expression was suppressed by peroxisome proliferator-activated receptor α (PPARα) activation and associated with impaired glucose homeostasis [15]. The let-7 microRNA family (a, d, and f) overexpression caused growth retardation and glucose intolerance, and impaired insulin sensitivity [15]. In contrast, the let-7 microRNA family knockout or anti-miR treatment protected mice from high-fat diet (HFD)-induced glucose intolerance and insulin resistance in muscle and liver. Recently, Yagai et al. further demonstrated that hepatocyte-specific let-7b/c2 knockout mice exhibited a pronounced resistance to obesity and fatty liver. Mechanism-wise, they revealed that let-7 microRNA inhibited the hepatic expression of PPARα and its heterodimer partner retinoid X receptor α (RXRα), therefore providing a negative feedback repression of PPARα signaling, which may act through a E3 ubiquitin ligase for RXRα, ring finger protein 8 (RNF8) [16].

***miR-149 promotes lipid accumulation.*** MiR-149-3p was identified in the EVs isolated from a human colorectal carcinoma cell line, HCT116 cells [17]. Upregulation of miR-149 has been observed in steatosis models both in vitro and in vivo. Fibroblast growth factor 21 (Fgf21) is one of the target genes of miR-149, which slightly increased intracellular lipid content in a human hepatoma cell line, HepG2 cells [18]. Overexpression of miR-149-5p aggravated uric acid-induced triglyceride (TG) accumulation in mouse hepatocyte AML12 cells, the effects of which were prevented by FGF21 overexpression [19]. In contrast, inhibiting miR-149-5p ameliorated the TG accumulation via an FGF21-dependent manner. PR domain containing 16 (PRDM16), a transcriptional cofactor that promotes brown adipogenesis, is another direct target of miR-149-3p. Intrasubcutaneous adipose tissue injections of anti-miR-149-3p lentiviral vectors ameliorated hepatic steatosis and whole-body insulin sensitivity by increasing Prdm16 gene expression, and suppressing proinflammatory genes [20]. 

***miR-21 is associated with insulin resistance and fatty liver.*** EVs derived from a normal human colon mucosal epithelial cell line, NCM460 cells, contained higher concentrations of miR-21 following the activation of the substance P/neurokinin-1 receptor signaling pathway, which plays significant roles in colonic inflammation, promoting proliferation and migration of colon epithelial cells [21]. How intestinal miR-21 respond to energy intake is still unknown, although hepatic miR-21 is affected by the metabolic status of the body. For example, HFD feeding led to a decrease in miR-21 expression in mouse liver [22]. Controversially, Rodrigues et al. found that miR-21 expression was increased in the liver and muscle in a fast-food diet-induced nonalcoholic steatohepatitis (NASH) mouse model [23]. The discrepancy may be attributed to the different diets and mouse models.

miR-21 is an upstream regulator of a number of genes related to lipid metabolism, including genes involved in lipid anabolism (such as fatty acid-binding protein 7, Fabp7) and catabolism (such as Pparα). miR-21 null mice were protected from HFD-induced insulin resistance and glucose intolerance. Liver-specific miR-21 knockout alleviated liver steatosis by suppressing genes involved in fatty acid uptake and de novo lipogenesis [24]. miR-21 knockout also protected mice from fast-food diet-induced metabolic syndrome and liver damage, including steatosis, inflammation, and fibrosis [23]. Intravenous injection of miR-21 antagomir improved insulin sensitivity and lowered levels of serum TG, total cholesterol [10], low-density lipoprotein cholesterol (LDL-CHO), and high-density lipoprotein cholesterol (HDL-CHO) in a streptozotocin-induced type 2 diabetes mellitus rat model, as is partially mediated by TIMP metallopeptidase inhibitor 3 (Timp3), a direct target of miR-21 [25]. In contrast, miR-21 increased intracellular phospholipids and TG, and enhanced gene expression of key enzymes in lipogenesis, such as fatty acid synthase (Fasn), acetyl-CoA carboxylase (Acaca), and fatty-acid binding protein 5 (Fabp5) in A549 cells or H1703 cancer cells [26]. However, miR-21 was also reported to attenuate lipid accumulation in lipopolysaccharide (LPS)-stimulated macrophages through inhibiting toll-like receptor 4 (TLR4)—nuclear factor kappa B (NF-ĸB) pathway [27]. These discrepancies may be due to the different cell models or in vitro study conditions.

***miR-30c and miR-130a suppress lipogenesis both in vitro and in vivo.*** In an adherent-invasive E. coli (AIEC) infected condition, the invaded human intestinal epithelial T84 cell-secreted EVs could be taken up by macrophages. The cargos carried miR-30c and miR-130a to recipient macrophages to suppress ATG5 and ATG16L1 expression so as to inhibit autophagy of AIEC [28]. Interestingly, both microRNAs are also involved in lipid metabolism. For instance, miR-30c suppressed hepatic lipid synthesis by targeting lysophosphatidyl glycerol acyltransferase 1 (Lpgat1), inhibited VLDL production via suppression of microsomal triglyceride transfer protein (Mtp), and improved hyperlipidemia and atherosclerosis in Apoe^−/−^ mice [29]. miR-130a/b overexpression suppressed genes related to inflammation (Tnfa) and lipogenesis (Acaca and Pparγ), while increased lipolytic gene expression, including Pparα and carnitine palmitoyltransferase 1a (Cpt1α) [30]. Similarly, porcine preadipocyte differentiation was inhibited by miR-130a via suppressing the target gene, Pparγ [31]. Consistently, hepatic exosomes from miR-130a-3p overexpressing mice suppressed lipid accumulation in the 3T3-L1 cell line and recipient mice by downregulating lipogenic genes Fasn and Pparγ [32].

***miR-200c-3p targets lipogenic transcription factors and enzymes.*** miR-200c-3p was highly expressed in EVs isolated from LPS-stimulated HCT-116 colon cancer cells or gastrointestinal cancer patients. miR-200c-3p suppressed the migration and invasion activity of the colorectal cancer cell line pretreated with LPS, and promoted cancer cell apoptosis [33]. One of the direct targets of miR-200b/c is Jun, which could enhance lipid accumulation through activating the lipogenic sterol regulatory element-binding transcription factor 1 (Srebp-1) gene expression. In HFD-fed mice, levels of miR-200b/c in the steatotic livers were reduced, while its downstream genes, Srebp-1 and Fasn, were increased [34].

***miR-33 suppresses HDL synthesis and cellular cholesterol efflux.*** As one of the differentially expressed microRNAs in EVs isolated from γ-aminobutyric acid (GABA)-treated human epithelial cell line Caco-2 cells, miR-33 caught attention because of its functions related to neural activity. Since the oral administration of GABA facilitated relaxation and improved sleep, yet GABA itself cannot permeate the blood–brain barrier (BBB), Inotsuka et al. postulated that GABA might be carried by intestinal EVs to pass the BBB [35]. It is also possible that EVs transport GABA, miR-33, and others to peripheral tissues to contribute the metabolic effects on glucose and lipid metabolism [36,37]. Overexpression of miR-33 reduced circulating HDL levels [38], while miR-33-deficient transgenic mice exhibited significantly higher serum HDL and CHO concentrations [39]. When HFD-fed mice received long-term silencing of miR-33, higher circulating TG concentrations and severer hepatic steatosis were developed. These effects might be mediated by the increased expression of nuclear transcription Y subunit γ (Nfyc), a predicted target gene of miR-33 and a transcriptional regulator for Acaca and Fasn [40]. Other confirmed targeting genes of miR-33 include adenosine triphosphate–binding cassette transporter (Abca1), which facilitates CHO efflux to apolipoprotein A1, and ATP binding cassette subfamily G member 1 (ABCG1), which mediates CHO efflux to newly generated HDL [38].

***miR-125 protects the body from steatosis and insulin resistance through its suppression of genes involved in lipogenesis.*** Exosomal miR-125a/b was reduced in glucagon-like peptide-2 (GLP-2)-treated rat residual jejunum tissue, and mediated the intestinotrophic effect of GLP-2 via elevating the levels of target molecule, myeloid cell leukemia-1 (MCL1) [41]. In genetic and diet-induced obese mouse models, hepatic miR-125a expression was decreased, whereas, miR-125a overexpression improved insulin sensitivity and alleviated hepatic steatosis in HFD-fed mice [42]. miR-125b overexpression in porcine adipocytes decreased lipid droplet accumulation and TG concentration in vitro by targeting stearoyl-CoA desaturase 1 (Scd1), which is an essential enzyme in lipogenesis that catalyzes the formation of monounsaturated fatty acids (MUFA) from saturated fatty acids [43]. Consistently, tail vein injection of miR-125b-expressing vector also reduced hepatic TG content [44]. Contrarily, HFD-fed miR-125b-2 knockout mice developed impaired glucose utilization and insulin sensitivity, and exhibited increased liver and white adipose tissue weights, which might have resulted from significantly elevated Scd1, Pparγ, or Cebpa expression [45]. In addition, fatty-acid elongase 6 (Elovl6), which facilitates the elongation of saturated fatty acids to C18, has been identified as a target of miR-125a and gene expression was suppressed due to miR-125a overexpression [42].

***miR-6769-5p is another intestinal EV microRNA found relevant to dyslipidemia.*** miR-6769-5p was found in the EVs collected from the Caco-2 cell culture medium, and was upregulated by carnosine treatment, which could enhance neurite growth via suppression of its target gene, ataxin 1 (Atxn1) [46]. miR-6769-5p might also participate in the regulation of lipid homeostasis, as it was elevated in serum samples from hyperlipidemia patients revealed by a microRNA microarray analysis. However, further studies are required to confirm the contribution of miR-6769-5p to lipid metabolic regulation [47].

***miR-29 aggravates lipid metabolism and impairs glucose uptake.*** miR-29 has been identified in human IEC line NCM460-derived exosomes and was upregulated upon toll-like receptor 3 (TLR3) activation [48]. miR-29 family suppression significantly reduced circulating CHO and TG in mice. Liver RNA-seq revealed that lipid synthesis pathways were downregulated, while the expression of anti-lipogenic molecules, sirtuin 1 (Sirt1), and aryl hydrocarbon receptor (Ahr), were upregulated. Consistently, in vitro radiolabeled acetate incorporation assays confirmed that inhibition of miR-29 reduced de novo fatty acid and CHO synthesis in human hepatoma cells [49]. Furthermore, using a liver-specific Sirt1 deletion model, Hung et al. demonstrated that the effect of miR-29 inhibition on hepatic TG suppression was in a Sirt1-dependent manner [50]. Moreover, miR-29a and miR-29c were increased in the skeletal muscles of type 2 diabetes patients. miR-29 impaired skeletal muscle glucose uptake and glycogen content via decreasing abundance of glucose transporter-4 (GLUT4), and reduced fatty-acid oxidation potentially through suppressing the expression of peroxisome proliferator-activated receptor γ coactivator-1α (Pgc1α) [51]. miR-29a was upregulated in palmitic acid (PA)-treated C2C12 myoblasts. The inhibition of miR-29a led to upregulation of its direct target, peroxisome proliferator-activated receptor δ (Pparδ), and therefore increased transportation of GLUT4 to the plasma membrane, and improved insulin-dependent glucose uptake. Overexpression of miR-29a increased UCP3 and reduced PDK4, FATP2, and LCAD protein and mRNA levels, which are involved in lipid metabolism regulation [52].

***dsDNA activates STING pathway.*** Intestine-derived EVs have been demonstrated to carry dsDNA, including mitochondrial DNA (mtDNA) and nuclear genomic DNA (nDNA), the release of which is significantly increased when IECs are injured, such as murine colitis and active human Crohn’s disease. These EV-carried dsDNA could activate the stimulator of interferon genes (STING) pathway in macrophages to further augment the intestinal inflammation [53]. Interestingly, independent of its innate immune function, STING regulates polyunsaturated fatty acid (PUFA) metabolism through inhibition of the fatty acid desaturase 2 (Fads2)-associated delta-6 desaturase (Δ6D) activity in both mouse embryonic fibroblast cells (MEFs) and liver. Absence of STING increases the levels of PUFAs, long-chain PUFAs, and derivatives from the omega-3 branch, which are known to improve glucose metabolism and protect humans and animals from cardiovascular diseases [54]. Furthermore, omega-3 fatty acids increase the expression of Ucp1, Pgc1α, and Prdm16 genes, thus promoting the browning/beiging of white adipose tissue [55].

### 2.2. EV-Proteins

***IL-1β plays a negative role in lipid metabolism.*** Gasdermin D (GSDMD) is a protein that increases in epithelial cells both in dextran sulfate sodium (DSS)-induced colitis mice and IBD patients. Bulek et al. found that GSDMD guided the release of EVs containing interleukin-1β (IL-1β), which promotes the aggravation of inflammatory pathologies [56]. It is part of the body’s defensive mechanism that cytokines (such as IL-1β) induce the lipoprotein release to bind toxic products [57]. Selective depletion of Kupffer cells in the liver significantly ameliorated hepatic steatosis, and reduced liver TG content and lipogenic gene expression as well as hepatic IL-1β levels. Conversely, overexpression of IL-1β in primary hepatocytes enhanced TG accumulation and fatty acid synthesis [58]. IL-1β also mediates the development of atherosclerosis. Postprandial hyperlipidemia induces IL-1β production in circulating monocytes, which later become foam cells. Mice deficient in IL-1 or the IL-1 receptor were less likely to develop atherosclerosis, as is consistent with the observations of patients who had IL-1β polymorphisms [57]. In contrast, IL-1 receptor antagonist-deficient (IL-1Ra^−/−^) mice, which had excess IL-1 signaling, were obesity-resistant without alterations in food intake and energy expenditure, but decreased lipase activity and serum insulin levels. Excess IL-1 signaling in IL-1Ra^−/−^ mice even reversed maturity-onset obesity induced by monosodium glutamate, which ablates cells in the hypothalamic arcuate nucleus that are involved in the metabolic regulation and leptin signaling [59]. These inconsistent conclusions may be due to the different transgenic mouse models applied in vivo studies and likely involved distinct compensative mechanisms following the gene knockouts.

***Intestinal EV-containing proteins respond to high caloric diet.*** Using a hypercaloric (high-fat/high-sucrose) diet-induced obese and prediabetic mouse model, a proteomics analysis of the small intestine-derived EVs showed a vast alteration in their protein contents. For example, proteins related to glycolysis and gluconeogenesis pathways were downregulated, while fatty acid uptake and β-oxidation proteins were upregulated, indicating a hypercaloric diet shifts energy provision from carbohydrates toward fat. Interestingly, the proteins involved in antioxidant defense mechanisms and products of lipid peroxidation were upregulated in intestinal EVs isolated from hypercaloric-diet mice, suggesting an activation of antioxidant response in the gut cells, which may also be true in many other cells that are challenged with a high-fat/high-sucrose diet. In addition, there was a significant inhibition of proteosome-related proteins, suggesting an impaired ability for counteracting the deleterious effects of oxidative stress following high-fat/high-sucrose diet. Proteins that upregulated in the prediabetic group included (1) type I acyl-CoA thioesterases (ACOT1-6), enzymes that drive FFA and CoA production by hydrolyzing fatty acyl-CoAs [60]; (2) GSTA1 and GSTA2, which detoxify electrophilic metabolites of xenobiotics intracellularly, and inactivate metabolites under oxidative stress [61]. At the same time, proteins that play roles in carbohydrate metabolism, tricarboxylic acid (TCA) cycle, and CHO homeostasis were decreased [62].

***High mobility group box 1 (Hmgb1) deficiency promotes intracellular lipid accumulation.*** Cryptosporidium enteric infection triggered upregulation of inflammatory genes in the liver and spleen, suggesting some interorgan mechanisms. In vitro study revealed that Cryptosporidium-infected murine IEC line (IEC4.1) released EVs activated the NF-ĸB pathway and increased proinflammatory gene expression in primary splenocytes. In addition, HMGB1 protein was upregulated in EVs derived from infected IEC4.1 cells, which induced Ifn-γ gene expression in recipient cells [63]. In vivo study validated the expression of Hmgb1 in IECs. Moreover, intestinal epithelial-specific knockout of Hmgb1 exhibited increased scavenger receptor class B type 1 (SR-B1, a receptor for HDL) and decreased apolipoprotein B48 (ApoB48) in IECs, leading to impaired lipid packaging and chylomicron formation. As a result, IEC Hmgb1-deficient mice showed lower serum TG and CHO concentrations but atypic lipid droplet accumulation in IECs. Furthermore, IEC Hmgb1-deficient mice were protected from high CHO and fructose-enriched diet-induced NASH [64]. Hepatocyte-specific Hmgb1 knockout downregulated genes related to fatty acid β-oxidation and upregulated endoplasmic reticulum (ER) stress markers. Hepatocyte Hmgb1 deficiency led to increased body weight and hepatic fat deposition, but did not affect glucose tolerance and energy expenditure [65]. Deletion of Hmgb1 gene in hepatocytes also enhanced Lxrα and Pparγ expression. In contrast, Hmgb1 overexpression in liver protected mice from HFD-induced hepatic steatosis [66]. 

***Annexin A1 (ANXA1) functions to mitigate lipid overload in various types of cells.*** ANXA1 was discovered in EVs isolated from human IEC line SK-CO15. Using IEC monolayers, it accelerated the wound healing process in the scratch-wound healing assay. Consistently, administration of ANXA1-containing nanoparticles to Anxa1^–/–^ mice decreased intestinal mucosal ulceration [67]. Annexin A1 is also widely involved in lipid metabolism in multiple cells and tissues. Anxa1^–/–^ mice had significantly elevated serum TG concentrations [68]. Anxa1 expression was increased in adipose tissue of overweight patients [69]. In a high-fat diet/streptozotocin-induced diabetic mouse model, Anxa1 ablation aggravated diabetic nephropathy, manifested by more severe intrarenal lipid accumulation. In high glucose plus palmitate acid-treated human proximal tubular epithelial cells, ANXA1 inhibition reduced Thr172 AMPK phosphorylation, resulting in decreased expression of Pparα and Cpt1b, therefore increased lipid accumulation, inflammation, and apoptosis [70]. ANXA1 plays a significant role in inhibiting the development of atherosclerosis via an anti-inflammatory reaction [71]. HDL and apolipoprotein A1 upregulated ANXA1 via ERK, p38 MAPK, AKT, and PKC signaling pathways in human umbilical vein endothelial cells (HUVECs), and further inhibited cell surface defensive markers and chemokine secretion, therefore inhibiting monocyte adhesion [72]. 

***Prostaglandin E2 (PGE2) might be involved in tuning of lipolysis.*** Deng et al., demonstrated that intestinal mucus-derived, exosome-like nanoparticles (IDENs) carried PGE2 and other lipid content that induced anergy of liver NKT cells. IDENs treatment inhibited expression of Ifn-γ and Il-4 in α-GalCer-activated NKT cells. Furthermore, intravenous injection of IDENs protected mice from concanavalin A-induced liver injury by lowing circulating proinflammatory cytokines IFN-γ and TNF-α, and reducing liver cell apoptosis [73]. From the perspective of lipid metabolism, mice lacking the PGE2 receptor EP4 (Ptger4^−/−^ mice) developed obesity without an increase in circulating inflammatory cytokines. When compared with controls, Ptger4^−/−^ mice had significantly higher respiratory quotient values despite similar food intake and oxygen consumption rates. The PGE2-EP4 axis facilitated lipolysis by inhibiting insulin signaling cAMP/PKA in adipocytes, and maintained basal lipolysis in white adipose tissue (WAT) [74].

### 2.3. EV-Signaling Pathways

***Intestinal EVs were found to stimulate Akt/mTOR pathway.*** EVs derived from human colon epithelial cell line NCM460 or colon cancer (CC) cell line SW620 promoted the migration of colon adenocarcinoma SW480, the effect of which was mediated by the activation of the epithelial–mesenchymal transition process and the Akt/mTOR signaling pathway [75]. Although there is yet no direct evidence, it is also likely that these intestinal cell-line-derived EVs would activate the Akt/mTOR pathway in many other cells or tissues. mTOR is a protein kinase that functions as the catalytic subunit of protein complexes mTORC1 and mTORC2, the major regulators of growth in animals. mTORC1 is a sensor of nutrient levels in the body, while mTORC2 responds to nutrient-induced signals such as insulin or growth factors [76]. mTORC1 promotes adipocyte differentiation and hepatic lipogenesis, partly via S6K1, PPARγ, and SREBPs, which had been well reviewed [77]. mTORC2 is essential for early adipogenesis when embryo stem cells develop into adipocytes, but not in mature fat tissues. As a component of mTORC2, Rictor knockout resulted in higher body and liver weights following HFD feeding [78], indicating that mTORC2 plays a key role in the regulating lipid homeostasis. In the adipose tissue, mTORC2 promotes glucose uptake and mediates the suppression of lipolysis upon insulin stimulation [79]. Adipose tissue-specific ablation of mTORC2 resulted in excessive lipid hydrolysis, ectopic TG accumulation, and subsequent glucose intolerance [79]. mTORC2 promotes insulin-stimulated lipogenesis in the liver, while mTORC2 deficiency led to suppressed hepatic lipid synthesis, uptake, and consumption [80].

***EGFR performs as a prolipogenic signal.*** Epidermal growth factor receptor (EGFR) was significantly higher in EVs isolated from colorectal tissues of DSS-induced colitis mice (IBD-EVs) when compared with healthy controls. Moreover, IBD-EVs induced cell proliferation via activating EGFR-ERK signaling pathway in murine NIH3T3 fibroblasts [81]. EGFR promotes lipogenesis in cancer cells, for instance, in glioblastoma cell lines, EGFR-enhanced SREBP-1 cleavage, and nuclear translocation via EGFR-PI3K-Akt pathway. Interestingly, substantial EGFR signaling was necessary for fatty acid synthetic inhibition-induced glioblastoma cell apoptosis [82]. In colon cancer cell lines, EGFR signaling increased lipid droplet density by activation of PI3K/mTOR pathway and loss of Foxo3. The lipids produced were utilized for cancer cell proliferation [83]. In the noncancerous state, HFD induced EGFR activation in mouse liver. EGFR inhibition by tyrosine kinase inhibitor PD153035 significantly alleviated hepatic steatosis and glucose intolerance induced by HFD feeding [84].

## 3. Gut Microbiota Derived EVs Participate in Inflammation Modulation and Lipid Metabolism

During the past two decades, gut microbiota has gained more attention due to its significant effects on many aspects of the body metabolism, such as host immunity, obesity, diabetes, fatty liver diseases, and energy expenditure [85]. The mediating factors between the gut microbiota and systematic effects are still under study, one of which is via secreting EVs. Cargoes of EVs produced by gut bacteria contain microbial DNA, cell signaling molecules, and bacterial membrane components; these are associated with glucose and lipid metabolism.

***Microbial DNA from gut bacterial EVs deteriorates insulin secretion and sensitivity.*** Compared with that of healthy controls, intestinal microbial DNA-containing EVs (mEVs) in obese patients and HFD-fed mice delivered significantly higher levels of microbial DNAs into the bloodstream and β cells [9]. 16s rRNA gene sequencing showed excessive microbial DNA accumulation in the liver, skeletal muscle, and adipose tissue in HFD mice, whereas they were rarely detected in these tissues from healthy controls [86]. Pancreatic islets, where large amounts of microbial DNAs accumulated, showed increased proinflammatory cytokine abundance and reduced insulin production. The V-set immunoglobulin-domain-containing 4 (Vsig4; a complement receptor of immunoglobulin family) positive macrophages protected the pancreatic islets from mEV infiltration. Tail vein injection of obese mEVs caused islet inflammation, inhibited insulin secretion, and impaired glucose tolerance in Vsig4^−/−^ mice, but not in WT control mice. Mechanism-wise, microbiota-derived bacterial DNA-containing EVs contributed to the development of islet inflammation and β cell dysfunction by activating the cGAS/STING signal [9]. Moreover, liver CRIg+ (complement receptor of the immunoglobulin superfamily) macrophages prevent the body from gut bacteria interference by filtering their products from circulation in a C3-dependent manner. CRIg null mice accumulated mEVs in skeletal muscle and epididymal fat with high levels of proinflammatory cytokines, and exhibited impaired glucose tolerance and insulin sensitivity. When pretreated by electroporation and DNase to eliminate DNA, HFD mEVs did not trigger tissue inflammation [86].

***TLR2 promotes inflammation and lipid accumulation.*** EVs derived from pathogenic species of mycobacteria, the M. bovis Bacillus Calmette Guérin (BCG) and Mycobacterium tuberculosis (Mtb) H37Rv, were enriched with putative TLR2 ligands, such as the 19-kDa lipoproteins, LprA and LprG. Cytokines and chemokines were increased in mouse macrophages treated with mycobacteria-derived EVs in a TLR2-dependent manner. Consistent with this, intratracheal injection of EVs from virulent mycobacteria led to a florid inflammatory response in the lungs of WT mice, while TLR2-knockout mice were protected from lung inflammation [87].

Concerning lipid metabolism, in vitro experiments showed that BCG infection of macrophages increased Pparγ expression in a TLR2-dependent fashion, and in turn, triggered lipid body biogenesis. Interaction between CD36 and TLR2 was necessary for the process of lipid body formation [88]. Persistent stimulation (16 h) of TLR-2, 3, and 4 on M2 macrophages in vitro upregulated the expression of 5- and 15- lipoxygenases (LO), enzymes that dioxygenate polyunsaturated fatty acids in lipids with a cis,cis-1,4- pentadiene into cell signaling agents [89]. Experimental results in animal models further confirmed the proadipogenic role of TLR2. Tlr2^−/−^ mice were protected from HFD-induced obesity, insulin resistance, and liver steatosis, without macrophage infiltration and inflammation in adipose tissue [90]. However, the interaction between TLR2 and LPS/TLR4 signaling pathways causes an opposing observation. Cao et al. found that TLR2-deficient offspring of mice with prenatal LPS exposure had significantly more severe hyperlipidemia compared with WT offspring, with compensative upregulation of the TLR4/Myd88 signaling pathway in the adipose tissue [91].

***Sphingosine 1-phosphate (S1P) is an important biologically active sphingolipid that helps maintain lipid metabolism homeostasis.*** When treated with enteropathogenic bacteria-secreted particles, colon cell line MC38 produced EVs that contain higher concentrations of S1P, CCL20, and PGE2. S1P-containing EVs enhanced tumor growth in a STAT3-dependent manner compared to EVs from control groups. CCL20 and PGE2 are involved in the recruitment process of Th17 cells [92]. S1P is exclusively formed by two isoforms of sphingosine kinases (SphKs), SphK1 and SphK2, and activates various signaling pathways via transmembrane G-protein coupled receptors (GPCRs), S1PR1-5 [93]. The majority of plasma S1P binds to apolipoprotein M (apoM), which mostly combines with HDL, and the remaining associates with albumin, LDLs, or VLDLs [94]. Interestingly, S1P contents increased in the subcutaneous WAT of obese diabetic patients, suggesting a connection between S1P and lipid metabolism [95]. When LDL receptor-deficient mice were fed with high CHO–Western diet and were treated with SKI-II (a sphingosine kinase 1 inhibitor that substantially reduces plasma S1P levels) for 16 weeks, their aortic atherosclerotic lesions increased, circulation TG levels decreased, without changing their body weight and plasma total/HDL CHO [96]. Likewise, FTY720 (a sphingosine-1-phosphate analogue) treatment showed a positive effect on insulin sensitivity and reduced monocyte gathering in adipose tissue from HFD obese mice [97].

***Peptidoglycan (PGN), a constituent of bacterial cell walls, impairs lipid metabolism.*** PGN could be delivered to circulation and remote tissues by bacterial EVs [98]. Kaparakis et al. demonstrated that PGN in EVs derived from Gram-negative mucosal pathogens triggered NOD1-dependent signaling and induced immune responses both in vitro and in vivo [10]. PGN could be taken up by IEC-18 and colon adenocarcinoma HT29-CL19A cells, and by mouse columnar epithelial cells after gavage in a TLR2-dependent manner. When confluent HT29-CL19A monolayers were exposed apically to PGN, EVs originating from their basolateral side carrying PGN were detected [98]. PGN might buffer SREBP activation in drosophila, since those fed on a mixture of sucrose and E. coli or E. cc bacteria exhibited the activation of the lipogenic protein SREBP in adipocytes and enterocytes, yet the effect of E. cc that released PGN in larger amounts was milder [99]. Nevertheless, PGN exerted prolipogenic influence in mice. Subcutaneous administration of PGN increased hepatic and serum TG concentrations and exacerbated liver inflammation and fibrosis in mice. Lipogenesis-related genes such as Pparγ and Srebp1 were upregulated in hepatocytes after PGN treatment, which stimulated the NOD2- NFκB-PPARγ signaling [100]. PGN also suppressed the beiging of white adipose tissue through promoting macrophage M1 polarization, which led to adipose tissue inflammation via activation of adipocyte TLR2 receptors [101].

***Lipoteichoic acid (LTA) is a cell membrane component of gram-positive bacteria that has a lipid-elevating effect.*** LTA is contained in EVs originating from Lacticaseibacillus rhamnosus JB-1 and can be endocytosed by both dendritic cells and IECs, as demonstrated by in vitro experiments and by BALB/c mice in vivo, respectively [11]. The internalization of LTA activated TLR2 and induced IL-10 expression in dendritic cells, resulting in mucosal tolerance of the bacterium [11]. LTA treatment increased serum TG and CHO in rats in a dose-dependent manner. The elevation of serum lipid levels was caused by enhanced hepatic lipolysis and de novo fatty acid synthesis, but was diminished by adrenergic inhibitor pretreatment, indicating that LTA-enhanced hepatic TG secretion was mediated by alpha-adrenergic receptors [102].

***LPS stimulates inflammation and inhibits lipid metabolism.*** LPS is the major component of Gram-negative bacterium cell wall [103] and could be carried by bacterial EVs, performing as potent stimulators of TLR4 that triggers host immune responses. Resistin is an adipocytokine that led to insulin resistance. LPS suppressed resistin expression through a pathway that involved TLR4, JNK, CHOP-10, and C/EBPα/PPARγ in 3T3-L1 adipocytes [104]. Fedorova Ev Fau-Fock et al. found decreased fatty acids oxidation and increased accumulation of lipid droplets in cytoplasm after LPS incubation, which led to less ATP production to provide energy for hormone secretion [105]. LPS injection in pigs decreased lipid metabolic gene expression, such as Acaca1, Fasn, Scd, and Ucp2, while the abundance of adiponectin and zinc-α2-glycoprotein was significantly increased [106].

## 4. Discussion

The gut functions to provide digestion juice and absorb nutrients from diets, and transfers chylomicrons and other lipoproteins through the enterocytes into circulation [107]. The enteroendocrine system (EES) plays crucial roles in energy metabolism by generating peptide hormones that influence appetite and satiety in the central nervous system, and by delivering regulatory signals to peripheral tissues and organs, such as sending EVs to the pancreas, liver, and adipose tissues to maintain glucose and lipid homeostasis [6] (Figure 1). Current studies on the functions of intestinal EVs and the roles of their cargo in glucose/lipid metabolism are not well connected, which results from a lack of direct evidences on the delivery of gut-derived EVs to peripheral tissues and how those EV-contents affect glucose/lipid metabolism in the body. Future studies are required to unveil the casual relationship between gut EVs and energy metabolism.

Of note, there are discrepancies between studies in terms of the impact of certain microRNAs on lipid metabolism; these discrepancies might arise from different animal or cell models that applied in those studies. In addition, microRNAs can have a large number of targets; it is possible that different microRNA levels trigger disparate molecular pathways and result in conflicting effects. Furthermore, microRNA deletion or overexpression in mouse or cell-culture models is unable to fully recapitulate the physiological or pathophysiological conditions in either healthy subjects or patients [108]. Consequently, there might be inconsistencies between basic science and clinical studies. Finally, in circulation, it is a challenge to distinguish the gut-derived EVs versus those originating from other tissues. Novel gut EV-specific modification tools are necessary for better appreciation of their roles in metabolism in vivo. Overall, the impacts of gut-derived messages transmitted by EVs deserve further investigation.

## Figures and Tables

**Figure 1 genes-13-01964-f001:**
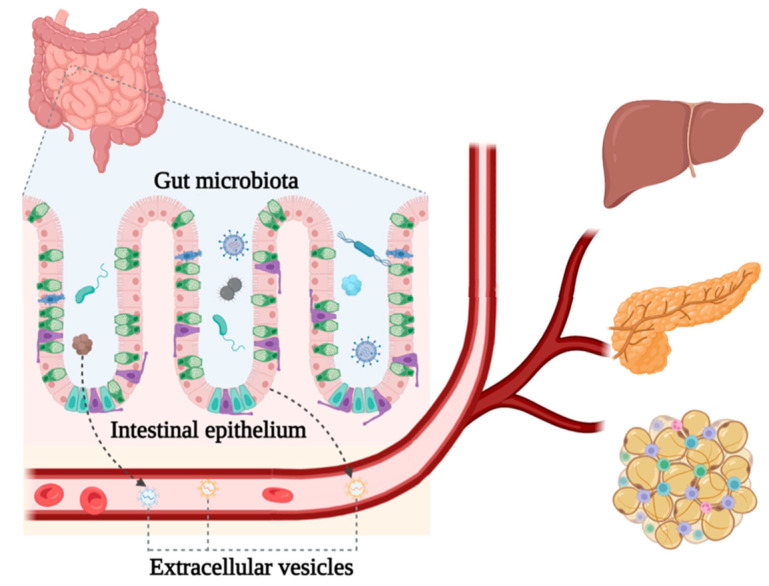
Gut-derived extracellular vesicles may participate in the regulation of lipid and glucose metabolism in remote organs. Extracellular vesicles originating from either intestinal epithelial cells or gut microbiota contain molecules that take part in the fine-tuning regulation of glucose and lipid homeostasis in liver, pancreases, and adipocytes.

## Data Availability

Not applicable.

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
