# Peer review of "Potential Mechanisms of Gut-Derived Extracellular Vesicle Participation in Glucose and Lipid Homeostasis"

_genes, 2022, doi:10.3390/genes13111964_

Round 1

Reviewer 1 Report

Gut-derived extracellular vesicles may contribute to glucose and lipid homeostasis

by Tiange Feng, Weizhen Zhang and Ziru Li

This review focuses on gut extracellular vesicles (EVs), on EV cargo molecules and their potential effects on glucose and lipid metabolism in liver, muscle, pancreas and adipose tissue. Although direct evidence for the intestinal EV-cargo’s involvement in glucose and lipid metabolism is lacking, the review points to potential mechanisms described in other contexts. The review has a dense collection of glucose/lipid metabolism signaling mechanisms implicating different molecules, which happened to be present in gut EVs as well. The article correctly concludes that these mechanistic studies present many challenges, such as comparing in vitro with in vivo experiments and the hurdles of separating gut EVs from other EVs.

Comments, suggestions, minor edits

Line 12: “The intestine participates in the regulation of glucose and lipid metabolism…”

Line 19: “…muscle, pancreas and adipose tissue.”

Line 23: “Extracellular vesicles (EVs) are a heterogeneous group of membrane-enclosed structures ranging from 30nm to 200nm in diameter…” Larger EVs may be even in the micrometer range! Citation 1 is from 2001. Please add a more recent publication also on EV nomenclature!

Line 43: “EVs produced by intestinal mucosal cells are mainly in form of exosomes…” Please provide citation for this statement!

Line 56: “there is still lack of direct evidence of how…”

Line 66: “…that are associated with glucose and lipid…”

Line 68: “Let-7 impairs glucose and lipid metabolism. In a recent study, Zhang et al. [14] found that EVs derived from normal intestinal organoids carried higher concentrations of let7c-5p…” They found higher concentrations of let7c-5p miRNA?

Line 74: “Let-7 family (a, d and f) overexpression…” Let-7 family (a, d and f) miRNA? Please specify.

Line 76: HFD, please write full name (high fat diet)

Line 80: “…and its heterodimer…”

Line 86: “…one of the target genes of miR-149…”

Line 88: aggravated uric acid-induced triglyceride (TG) accumulation?

Line 101: “…respond to energy intake, although hepatic miR-21 is affected by the metabolic status of the body.”

Line 105: “…may be attributed…”

Line 126: “…could be taken up by macrophages.”

Line 256: “…homeostasis were decreased [62].”

Line 261: “Although there is yet no direct evidence, it is also likely that these intestinal…”

Line 324: “…cell surface defensive markers and…”

Line 327: “…carried PGE2 and others…” Other PGs or other molecules? Please specify.

Line 337: WAT: please write full name at first mention, white adipose tissue

Line 398: “…and the remaining associates with albumin, LDLs, or VLDLs [94].”

Line 465: “Figure 1. Gut-derived extracellular vesicles may participate in the regulation of lipid and glucose metabolism in remote organs.”

Line 60-337: “2. Gastrointestinal EV-containing substances are associated with glucose and lipid metabolism.” Please try to organize section 2 better (i.e. EV-nucleic acids (RNA, DNA), EV-proteins, EV-lipids; EV-signaling pathways, etc)

Because microRNAs may have a high number of targets (sometimes over 100 different targets), it is impossible to dissect their specific effects, the mechanisms involved and the interactions between the different pathways. Any minute change in one factor in this amazingly complex network could lead to a completely different outcome. Please comment on this in chapter 2.

Author Response

We appreciate the careful and thoughtful comments from Reviewer 1. We have gone through all the suggested edits and made changes/corrections, correspondingly. We have reorganized section 2 with subtitles as: 2.1 EV-nucleic acids (RNA, DNA); 2.2 EV-proteins; 2.3 EV-signaling pathways. Reviewer's insightful comment about microRNA is much appreciated, and has been added into section 2.1 and discussion.

Reviewer 2 Report

This manuscript summarizes the recent knowledge of cargo substances detected in intestinal extracellular vesicles, and connects these molecules with the regulation of glucose and lipid metabolism in liver, muscle, pancreases and adipose tissue. 

The literature cited and discussed in this paper is relatively comprehensive, and the content organization structure is clear.

The Discussion section is fairly simple and short, should be more in-depth analysis based on the research progress of the review .

Author Response

We feel encouraged by the positive comments from Reviewer 2 and expanded more in the discussion section. 
